# Low-Cost Efficient Wireless Intelligent Sensor (LEWIS) for Research and Education

**DOI:** 10.3390/s24165308

**Published:** 2024-08-16

**Authors:** Mahsa Sanei, Solomon Atcitty, Fernando Moreu

**Affiliations:** 1Department of Civil, Construction and Environmental Engineering, University of New Mexico, Albuquerque, NM 87131, USA; msanei@unm.edu; 2Department of Mechanical Engineering, University of New Mexico, Albuquerque, NM 87131, USA; satcitty4@unm.edu

**Keywords:** low-cost sensor, data collection, real-time monitoring, measurement, education

## Abstract

Sensors have recently become valuable tools in engineering, providing real-time data for monitoring structures and the environment. They are also emerging as new tools in education and training, offering learners real-time information to reinforce their understanding of engineering concepts. However, sensing technology’s complexity, costs, fabrication and implementation challenges often hinder engineers’ exploration. Simplifying these aspects could make sensors more accessible to engineering students. In this study, the researcher developed, fabricated, and tested an efficient low-cost wireless intelligent sensor aimed at education and research, named LEWIS1. This paper describes the hardware and software architecture of the first prototype and their use, as well as the proposed new versions, LEWIS1-β and LEWIS1-γ, which simplify both hardware and software. The capabilities of the proposed sensor are compared with those of an accurate commercial PCB sensor. This paper also demonstrates examples of outreach efforts and suggests the adoption of the newer versions of LEWIS1 as tools for education and research. The authors also investigated the number of activities and sensor-building workshops that have been conducted since 2015 using the LEWIS sensor, showing an increasing trend in the excitement of people from various professions to participate and learn sensor fabrication.

## 1. Introduction

The field of Structural Health Monitoring (SHM) is currently undergoing a transformation, driven by advancements in data collection and analysis techniques. SHM uses algorithms, resources, and different tools to send recurring evaluations of the integrity and safety of civil infrastructure [1]. Assessment of the structures health status is necessary to diminish repair costs and maintenance and ultimately enhance infrastructure safety [2]. In addition, SHM assists civil engineers in keeping track of structural information and making informed decisions [3]. In the past, civil infrastructure evaluation has been a domain of manual, time-consuming, and subjective evaluations [4]. However, with the increasing need for better assessment tools, researchers have been looking for innovative solutions such as sensors [5].

Sensors are one of the tools that are extensively utilized in data collection for SHM systems to quantify structural responses over time (displacement, strain, rotation, temperature, humidity, acceleration, etc.) and consequently identify any unusual issues that can have detrimental effects, such as structural damage [6]. Concepts of “smart cities” have been established as a way to use technology-based solutions to alleviate infrastructure issues and enhance infrastructure performance [7]. Smart cities with sensors and a real-time diagnostics system have contributed to this rapid advance change with the integration of high-quality technology into the life of structures [8].The application of smart technology such as sensors for infrastructure systems has been identified a solution which highlights new responsibilities for civil engineers [9]. However, the widespread use of sensors in SHM brings its own set of challenges and opportunities that need to be explored further.

According to the American Society of Civil Engineers (ASCE) 2025 vision, engineers rely on using real-time access to data, sensors, analytical tools, and other sophisticated technology to help them make informed decisions [10]. This strategic plan also demonstrates the worldwide vision for the civil engineering future and delineates the importance of engaging more people in the field of civil engineering. Hence, in the future vision of ASCE, there should be a plan that not only emphasizes widespread use of sensors but also transfers the knowledge and application of this technology to the new generation in a simple way. Therefore, the knowledge of sensors should be brought up in a broader context to be transferable and applicable in a variety of applications.

The demand for effective and affordable monitoring systems such as sensors keeps rising. The low-cost sensors provide a number of advantages over traditional instrumentation, including a lower unit price, a smaller size, portability, and the capacity to capture acceleration in three different axes [11]. To date, different researchers have used or developed low-cost sensors in engineering applications. Due to the rapid development of commercial wireless and mobile computer technologies, the hardware capabilities of sensing unit designs continue to advance [9]. Hence, developing a low-cost sensor with simple fabrication can be very useful for health monitoring of structures.

Research in academia is expanding on the use of low-cost sensor technology in different areas. As the price of sensor equipment has decreased in recent years, the variety of applications for information sensing technology has significantly increased [12]. For instance, Weng et al. developed and validated a real-time health monitoring system using integrated wireless network system for civil constructions. The suggested system allows for the simultaneous collection and analysis of data from a number of wireless sensing units that can operate as a set of analog sensors in real-time. The sensor signals are enhanced by incorporating low-cost signal conditioning circuits. Extensive lab and field tests support the viability and dependability of this integrated wireless SHM network system [13]. In another study, the researcher instrumented a structure by dispersing several synchronized low-cost LIS344ALH accelerometers alongside to develop a system capable of obtaining the structural modal. Although the proposed system of synchronized sampling from different nodes had comparable results to piezoelectric sensors, this system needs a cloud and high computation effort for implementation. The performance of an ADXL335 MEMS low-cost accelerometer was evaluated with analog output for bridge vibration measurement. This accelerometer was tested under ten different harmonic excitations and on a typical highway bridge [14]. The results were compared with an instrument-grade accelerometer that showed this developed system could identify the dynamic characteristics of a structure. They only considered the z-direction to compare the output, as this direction had the worst case in terms of noise density [15]. Other group of researcher proposed a new data acquisition system by combining five low-cost accelerometers which show high accuracy on low frequency and low acceleration amplitudes [16]. Even though this Cost Hyper Efficient Arduino Product (CHEAP) system has a very low error at the frequency between 0.5 and 10 Hz, the fabrication of this sensor is not easy, depends on the computer, and has high noise density. It also measures the acceleration in one direction. Hence, they upgraded their system and developed a new Low-cost Adaptable Reliable Accelerometer (LARA) using Arduino technology that can be used in SHM [17]. They tested the system on a short-span Foot Bridge and compared the result with a high-precision commercial sensor. Previously, researchers developed a group of Low-Cost Efficient Wireless Intelligent Sensors (LEWIS) to provide a straightforward and affordable platform for SHM sensing and introducing new users to the world of sensors [18]. However, in the SHM, it is important to have sensors that offer benefits such as affordability, ease of manufacturing, and high accuracy at the same time for effective data collection.

The primary objective of this paper is to introduce and explore three generations of low-cost sensors called LEWIS that can be used for data collection required in structural health monitoring. These sensors are low-cost and simple enough for engineers without expertise in electrical or computer engineering to fabricate and use them in engineering. Importantly, this accessibility does not come at the expense of accuracy. This paper delves into hardware specifications, cost considerations, and the fabrication process of these sensors. It also summarizes the improvements and comparative testing of this low-cost sensor and a commercial sensor as a ground truth. Furthermore, the authors present their experience of conducting outreach activities, classes, and workshops with LEWIS sensors from 2015 until fall 2023. The contents of this work enable other educators and researchers to use this platform in their institutions for multiple educational and outreach activities.

## 2. Challenges and Benefits of Sensor Technology in SHM

In recent years, structural health monitoring and inspection has become increasingly important due to the aging of the infrastructure [19]. The lack of widespread use of sensors for data collection and structural monitoring can be attributed to several factors. Firstly, traditional methods of data collection and analysis often require extensive manual labor. Fabrication of sensors is also time consuming which makes it difficult for engineers and educators to incorporate sensor technology into their application of interest. Additionally, sensor technology has been expensive and often out of reach for many engineering fields. Even though engineers are eager to introduce new technologies for data collection, they often find the idea of building a sensor to be challenging and complex. However, low-cost sensors have lowered the barriers for engineers and educators, making this technology more accessible and easier to use.

The traditional wired monitoring systems usually present a very complex setup in terms of connections with multiple wires to the sensors, data acquisition units, and data transmission modules [20]. Hardware procurement is expensive, in addition to maintenance costs for such labor-intensive systems. Therefore, it becomes somewhat complicated and is less adaptive to a dynamic kind of monitoring demand or to changes in the environment. [21]. For instance, large bridges demand long stretches of wiring through the structure for installing conventional SHM; this turns out to be not only costly but also time consuming. Maintenance of these systems demands constant inspection of the wiring and sensors in order to cut down on costs [17].

In contrast, the modern low-cost sensor systems, such as the use of Arduino platforms, are much more effective and cost efficient. These consolidate all the required components on a single compact platform, which thereby reduces the complexity and cost of both installation and maintenance [22]. The ease of customizing and scaling the Arduino using a versatile and supported microcontroller makes the system very crucial in aiding to reduce these costs. This novel approach has a double advantage—it reduces initial costs and minimizes long-term expenditure associated with maintaining and upgrading the system [23]. For instance, an Arduino-based SHM system may be well implemented with minor wiring in different kinds of structures. Besides that, any change that is needed can be completed through easy reprogramming without a requirement for large modifications in the hardware.

Modern low-cost sensor systems are particularly beneficial for use as the access point for engineers and educators who could previously not afford to use traditional sensor systems, primarily because of cost and complexity [24]. They can now do so to a great advantage in advancing their projects and research. This quick access to technology further simplifies the ability for small engineering firms or academic institutions, working with a limited budget, to embrace effective SHM solutions that improve safety and longevity for a wider array of structures.

The next section introduces new low-cost sensors that have advantages in the civil engineering field. This sensor is simple to fabricate, even for engineers without prior programming knowledge, allowing anyone to execute the code and obtain results for environmental sensing measurements. This sensor is also low cost and affordable for engineers and researchers, making it accessible to a wider range of individuals.

## 3. Prototype Design of LEWIS1

This section provides information about a simplified motion sensor that has been developed and used for engineering purposes and educational activity. In this section, the hardware, software, and characteristics of this sensor named LEWIS1 will be explained.

### 3.1. LEWIS1

The Low-cost Efficient Wireless Intelligent Sensor, called LEWIS, is the first generation of the low-cost sensing series that aims to resolve issues within Structural Health Monitoring (SHM), assist in the decisions of infrastructure managers, and educate individuals on the concept of low-cost sensing. The simplistic architecture of the LEWIS sensor makes it low cost and easy to fabricate, even for a person without prior knowledge of electric and computer engineering. The components of this low-cost sensing platform are described in the hardware section. Next, the Arduino platform, which provides the software needed for capturing acceleration data, is introduced. Finally, the limitations and the ways for improvement are discussed.

#### 3.1.1. LEWIS1 Hardware

The LEWIS1 unit is composed of three main parts, the microcontroller, sensing element, and development shield. The microcontroller is an ATmega328p-based Uno R3 board (Arduino, Ivrea, Italy) called Arduino with 14 digital I/O pins. A USB cable or an external power source can be used to power the Arduino board (such as an AD-to-DC adapter or a battery). A tri-axial MMA8451 digital accelerometer (Adafruit LLC, New York City, NY, USA) with a built-in 14 bit ADC serves as the sensing component [25]. Its usage spans a wide spectrum, from ±2 g up to ±8 g. The last part is the development shield which includes a breadboard that turns the Arduino and accelerometer into a circuit by carrying all Arduino pins to the upper layer. Figure 1 shows the main components of the LEWIS1 and the wiring of the accelerometer to Arduino board.

Additionally, the researchers massed LEWIS1 building kits to further simplify the sensor assembly process. The building kit was comprised of the LEWIS1 components, a box casing to house the components, and access to an assembly and operations manual.

Table 1 shows the cost of the LEWIS1 components. According to this table, the total cost of the LEWIS1 kit is less than USD 60 which demonstrates the cost effectiveness of this sensor.

#### 3.1.2. LEWIS1 Software

The Arduino integrated development environment (IDE) is a simple and easy-to-learn software for writing Arduino programs, also known as Arduino sketches. It is possible to write, compile, and deploy code on a board using IDE. This IDE supports a C/C++ dialect that employs unique code organization guidelines [26]. The researchers developed an Arduino sketch that allows the Arduino IDE software (version 1.8.4 or higher) to read incoming sensor data. For this sketch, there are two libraries for this particular accelerometer named “Adafruit MMA 8451 Library” and “Adafruit Unified Sensor” that are required to be installed. The code used for collecting data using the LEWIS sensor in this study is available at https://github.com/satcitty/lewis (accessed on 23 July 2024). Once the sensor A/B USB cable is plugged into the computer, the provided Arduino sketch can be uploaded onto the Arduino board, which enables the user to start reading and logging the acceleration in three different axes. The data are written in a .txt file and can be saved for further analysis. Figure 2 shows the system that is used for data collection with the LEWIS1 sensor.

### 3.2. Limitations and Improvement

The LEWIS1 capabilities bring new opportunities for engineers and educators to perform experiments with their own built sensor without having extensive knowledge of electrical components. The researchers trained educators and new learners to build LEWIS1s. The integration of sensor technology into the engineering education curriculum is an effective way to enhance students’ learning experiences. By incorporating sensors into the curriculum, students can gain practical skills in real-time data collection and analysis, which are essential in problem solving and decision making. It can help students develop critical thinking skills as they interpret and analyze data, promoting a deeper understanding of concepts and theories. Additionally, students can gain hands-on experience in data collection and analysis, which is an essential skill in many fields.

The LEWIS1 sensor is an inexpensive tool particularly developed for the needs of education, research, and potential engineering application. The activities designed by this research group are for two distinct categories: educational activities and research activities. The educational activities are designed for K12th and college students, aiming to provide hands-on learning experiences that make complicated concepts easier to understand. On the other hand, our research activities are designed for professionals and researchers, including sensor workshops and also teaching advanced versions of the LEWIS sensor, which are beyond the scope of this study. Feedback from both educational and research participants has shown several needs for the LEWIS sensor to be more user friendly and effective:Participants showed an interest in a faster and more efficient fabrication process. This includes reducing the time required to put together the sensor, making it more convenient for use in projects and workshops.The LEWIS1 currently needs the accelerometer to be soldered to the header pin, which is challenging for many engineers and students. Hence, the design should incorporate snap-fit components, or pre-soldered parts that can be easily assembled. Additionally, the four wires that need to be connected to the header pins can be replaced by one wire with four outputs (1 to 4), simplifying the connection process.Expanding the sensor’s capabilities to measure other degrees of freedom, such as angular velocity and temperature, would make it more versatile and useful for a wider range of applications.Using available and affordable components can make the sensor more accessible to a broader audience.

Incorporating these improvements based on participant feedback will make the LEWIS1 sensor user friendly, and effective for both educational and research applications. The next section will introduce and summarize the use of a new accelerometer that could be an alternative to the current LEWIS1.

## 4. Prototype Design of Upgraded LEWIS1

To enhance the potential capacity of the LEWIS1, a new iteration of LEWIS1 can be developed to address some of the challenges of the present version and improve it. This section introduces the upgrade of LEWIS1 based on the limitations and challenges encountered during educational and research activities. The hardware, software, and characteristics of two proposed sensors, LEWIS1-β and LEWIS1-γ, will be discussed. Additionally, the characteristics of these different versions will be compared at the end of this section.

### 4.1. LEWIS1 Upgrade

The LEWIS1 is capable of measuring tri-axis acceleration with a low-cost, portable, small-size platform. Different accelerometers can be replaced to build a new platform for measuring acceleration. The authors propose a new version of this device, named LEWIS1-β, which presents several advantages over the previous model, improving both functionality and user experience. The new platform architecture is improved by replacing the accelerometer, providing simpler assemblage, and using simpler software. One of the most important differences between these two sensors is that the LEWIS1-β not only measures the tri-acceleration but also has the capability of measuring the temperature and the angular velocity. The introduction of the new sensor, LEWIS1-β, will increase its potential applications and simplify its fabrication process due to fewer components. Additionally, the authors have developed the LEWIS1-γ variation, which has an enclosure redesign that facilitates the accelerometer’s easy insertion while still using the same accelerometer as the LEWIS1-β. This model maintains the same functionality while reducing component count and simplifying the manufacturing process. Further details regarding the software, hardware, and user interface of these new sensors will be discussed in the following sections.

#### 4.1.1. LEWIS1-β and LEWIS1-γ Hardware

The newly developed LEWIS1 is comprised of two primary components, a microcontroller, and a sensing element. The microcontroller board is an Arduino Uno SMD R3 based on the ATmega328p single-chip microcontroller, which has 14 digital I/O pins that are utilized in the new sensor platform. The new sensing element is the LSM6DSOX (Adafruit LLC, New York City, NY, USA), which is a six-degree of freedom (DOF) inertial measurement unit (IMU) that includes an accelerometer and gyroscope [27]. The accelerometer has a decent range of ±2/±4/±8/±16 g at a 1.6 Hz to 6.7 kHz update rate. Figure 3 and Figure 4 show all the required components for the assembly of LEWIS1-β and LEWIS1-γ and the required wiring for connecting the accelerometer to the Arduino board. It should be noted that both LEWIS1-β and LEWIS1-γ use the same accelerometer, and the wiring is shown in Figure 3.

Table 2 shows the cost of the LEWIS1-β and LEWIS1-γ components. The only difference between these two versions of the sensor is the enclosure and accelerometer holder. This table shows that even though this sensor has more capability compared to the LEWIS1, the total cost of a kit containing all the components is less than USD 60.

Researchers chose Arduino due to its adaptability and simplicity in linking with additional sensors like accelerometers, gyroscopes, and magnetometers, as well as its widespread market availability for the end user. Additionally, Arduino features a built-in development open-source platform that supports the C programming language. The term “Open Source” denotes that all the board’s resources, including the design and CAD files, are free and available to everyone [28]. This enables users to freely tune the microcontroller’s performance through a USB connection. This implies that anyone can change it to suit their needs. With Arduino, both professionals and students may easily and affordably build microcontroller computers that can communicate with their surroundings.

To assemble all the components of LEWIS1-β together and enclose them in the box, a holder plate is designed to further simplify the assembly process. The holder is a small plastic plate that securely holds the sensing element. The holder plate is designed to consider the lack of a breadboard in the LEWIS1-β sensor. The plate is composed of a central rectangular part and three connected arms. The central part is the housing location of the sensing element. This part utilizes two snap clips along with two small protruding cylinders to effortlessly secure the sensing element to the plate. Each arm contains a 3.5 mm hole at the very end which is used to affix the plate to the new LEWIS1-β enclosure. Figure 5 demonstrates the top, isometric view and real picture of the designed plate and accelerometer. Figure 6 shows the 3D-printed enclosure of LEWIS1-γ, which has a snap-in design for holding the accelerometer inside.

#### 4.1.2. LEWIS1-β and LEWIS1-γ Software

The LEWIS1-β and LEWIS1-γ sensors are both compatible with the Arduino IDE. They require an Arduino sketch for data processing. To read and plot the data, users must install three specific libraries named “Adafruit Unified Sensor”, “Adafruit BUSIO”, and “Adafruit LSM6DS”. Once these libraries are installed, users will be able to connect to the port, upload the code on Arduino, log data, visualize, and record the data seamlessly.

### 4.2. LEWIS1 and LEWIS1-β Assembly Comparison

The assemblage of the new sensors has some differences from the LEWIS1. First, the LEWIS1-β sensor does not need a breadboard and shield since the accelerometer is directly mounted on the 3D-printed plate. Second, without a breadboard, the soldering of header pins to the accelerometer would not be required. Third, instead of four single wires, a four-pin header jumper wire is used, which reduces the time and complexity of the wiring. Figure 7 shows the assembly of the three LEWIS sensors.

This reduction in component count can lead to several benefits, such as lower manufacturing time, improved reliability, and increased ease of assembly. Additionally, the fact that the LEWIS1-β and LEWIS1-γ sensor require only one wire connection instead of four may simplify the wiring process and reduce the risk of errors. Table 3 shows the number of components for LEWIS1, LEWIS1-β and LEWIS1-γ.

Figure 8 shows the number of components required for assembling a sensor of each type. According to the figure, the LEWIS1 assembly is more complex as it needs four different wires to be connected to the breadboard using four separate header pins, while LEWIS1-β uses a four-to-one wire which reduces the wiring complexity, and it does not need a header pin as they are mounted on the Arduino themselves. LEWIS1-γ not only uses a four-to-one wire, but also removes the need of accelerometer holder.

Table 4 summarizes the properties of the three version of the LEWIS1 sensors. The LEWIS1-β has been improved in many aspects including the reduction in the number of components which makes the fabrication simpler and faster while maintaining almost the same cost. It has a wider range of acceleration usage and can be also used for calculating angular velocity. One factor that makes the fabrication of LEWIS1-β harder than the LEWIS1 is that it has a component that needs to be 3D printed for holding the accelerometer while the LEWIS1 only requires soldering. Generally, both accelerometers are easy and quick to fabricate, especially for engineers without any background of electrical and computer engineering, and they also provide accuracy comparable to a commercial sensor.

Successful SHM requires specific and targeted sensor needs. An accelerometer measuring up to ±2 g is usually sufficient in this case since it records the low-frequency vibrations and accelerations required for structural analyses. In general, applications requiring SHM that monitor small structural motions and vibrations do not need higher values like ±4 g or ±16 g. Similarly, gyroscopes typically need a ±250 dps range. Higher values, such as ±1000 dps or ±2000 dps, are usually not needed for SHM because it focuses on relatively slow rotating movements. Thus, by selecting for lower ranges and resolutions, one can improve performance and cost effectiveness by selecting sensors that have matching monitoring needs for SHM.

Table 5 presents a comparison of three accelerometer types that meet different requirements for SHM applications: LEWIS1, LEWIS1-β, and LEWIS1-γ. In addition to practical concerns like verified uses in various bridge types and ease of installation, the requirements cover technical standards like acceleration range, resolution, and sampling rate. Every need is assigned a number between 1 and 3, where 3 denotes the best possible compatibility or performance for a given model. For each model, a total SHM score is computed at the bottom; LEWIS1-β had the highest score 19, followed by LEWIS1-γ 17 and LEWIS1-14. A rapid evaluation of the models’ overall suitability for SHM applications is made possible by this rating system. It is crucial to remember that research and education are the primary uses of these sensors. In particular, more time is required to thoroughly examine the capabilities of LEWIS1-γ, and future testing is anticipated to improve its score. Though this possibility has not yet been completely tested, the authors’ experience shows that these sensors might also be utilized for buildings and other structures, even though the table focuses on bridge applications.

### 4.3. LEWIS1 and LEWIS-β Accuracy Comparison

In addition to the assembly comparison of the developed sensor, the accuracy of both accelerometers used in the LEWIS needs to be compared to a commercial accelerometer. To assess the performance quality of the two sensors in comparison to the commercial accurate sensor, we calculate two indices. The first is the root-mean-square error (RMSE), which quantifies the deviation of the measurements from the LEWIS sensors and the reference accelerometer from the true values recorded by the commercial sensor. Since the sampling rates of the LEWIS sensors and the reference accelerometer differ, for a fair comparison and analysis, it is necessary to resample the data to the LEWIS sampling rate of 100 Hz. The following equation is employed to compute the RMSE.
(1)$$RMSE=\sqrt{\frac{\sum_{i=1}^{n}{\left({acc}_{PCB}-{acc}_{LEWIS}\right)}^{2}}{n}},$$
where,

${acc}_{PCB}$ and ${acc}_{LEWIS}$ are the acceleration of commercial and LEWIS accelerometers, respectively. n represents the number of data points. The percentage difference is calculated based on the following equation.
(2)$$Percentage\;difference\left(\%\right)=\frac{\left|{acc}_{PCB}-{acc}_{LEWIS}\right|}{\frac{{acc}_{PCB}-{acc}_{LEWIS}}{2}}*100$$

The second index for comparing two signals is signal-to-noise ratio (SNR). SNR represents which sensor has the most accurate response. The SNR can be calculated using Equation (3).
(3)$$SNR=10\;\log\;\left(\frac{Signal\;power}{Noise\;power}\right)$$

The details of the experiment and analysis of the result will be explained in the next section.

## 5. Experimental Evaluation and Validation

Researchers conducted an experiment to compare the capabilities of the LEWIS1 platform with the LEWIS1-β. Since both LEWIS1-β and LEWIS1-γ use the same accelerometer, only LEWIS1-β will be tested in this experiment. A commercial reference accelerometer was used as the ground truth to verify the accuracy and effectiveness of the proposed measurement platform. The primary measurement axis for data collection was the z-axis. The excitation inputs and performance evaluation standards used to test and evaluate the suggested sensing platform are outlined below. The final section examines the data gathered from the experiments and makes comparisons between the suggested system, LEWIS1, and the commercial sensor measurement.

### 5.1. Experiment Setup Description

In this study, we compare the LEWIS1 and the newly suggested LEWIS1-β sensing platforms with a uniaxial #353B33 ICP accelerometer—DC type (PCB Piezotronics Inc., Depaw, NY, USA), a capacitive sensor. Two sensors are mounted on an APS 113 Shaker (APS Dynamics, Inc., San Juan Capistrano, CA, USA), capable of providing up to 133 N peak sine force with a maximum stroke of 158 mm. Both LEWIS sensors are connected to a VibPilot DAQ system (m+p international, Hanover, Germany), an eight-channel data acquisition system manufactured by M+P International. The VibPilot features a 24 bit A/D converter with anti-aliasing sampling rates of up to 102.4 kHz.

Each LEWIS sensor is connected to a computer for data collection using the IDE application, while the PCB sensor is connected to the DAQ system for data reading and recording. Consequently, one channel of the VibPilot is dedicated to the PCB sensor as a response, and another channel is used to connect the smart shaker as an excitation input. Figure 9 illustrates the entire system’s connection to the DAQ platform.

To be able to assemble all the three sensors together and attach them to the smart shaker, a plate was designed and 3D printed. This component secures the entire system, ensuring that the same excitations are simultaneously applied to all three sensors. The collected data will be saved and further analyzed using MATLAB software (2021.b and higher). Figure 10 shows the experimental setup, including the shake table, the two LEWIS sensors (MMA8451 and LSM6DS), and the commercial PCB accelerometer.

### 5.2. Excitation Input

The two sensors and the PCB accelerometer are mounted on the smart shaker, which is connected to VibPilot to excite all three sensors simultaneously. Although the LEWIS sensor is capable of measuring acceleration in three axes, in this experiment, the vibration is only measured in the z-direction. In the future, other directions will be tested. We select a total of six different excitations as inputs to the system: (1) 1 Hz Sinusoidal excitation, (2) 3 Hz Sinusoidal excitation, (3) 5 Hz Sinusoidal excitation, (4) 10 Hz Sinusoidal excitation, (5) Sine sweep excitation with a frequency range of 0–10 Hz, and (6) Band-limited White Noise (BLWN) with a cap of 10 Hz. In all excitation scenarios, we capture readings from the LEWIS accelerometers and the PCB sensor at 100 Hz and 1024 Hz, respectively. For each stimulation case, we gather measurement data for 16 s. Table 6 presents the list of excitations.

### 5.3. Result and Analysis

Researchers applied six different excitations to the shake table, simultaneously exciting the LEWIS1, LEWIS1-β, and PCB sensors mounted on the plate. The LEWIS sensors captured data at a 100 Hz sampling rate, with the data being logged in the connected computer for each sensor. The VibPilot generated responses and delivered excitations to the smart shaker. The response of the PCB accelerometer, sampled at 1024 Hz, was recorded by the VibPilot.

It is important to note that in order to synchronize the initiation and termination of the test for the LEWIS sensors, some minor discrepancies in the time vectors were unavoidable. To mitigate the effects of abrupt movements and noise at the beginning and end of the excitation period, the first and last two seconds of the response data were removed. Consequently, the experiment was analyzed over a 12 s duration.

Figure 11 shows the time domain responses and frequency domain response focusing on power spectral density (PSD) of both sensors, along with the PCB accelerometer, under various excitation conditions. The PSD plots show the dominant frequencies which matches the excitation input, and the frequency obtained from the PCB sensor. Notably, the figure includes a zoomed-in plot representation of the sinusoidal excitation at 5 and 10 Hz for the time domain. This zoomed-in plot, spanning the time interval of 4 to 6 s, has been included to enhance the clarity and detail of the displayed data.

Table 7 shows that the RMSE of LEWIS1 is higher than that of LEWIS1-β, indicating that the performance of LEWIS1-β is superior to LEWIS1. Additionally, Table 7 displays the signal-to-noise ratio (SNR) for both LEWIS sensors and the PCB sensor. Larger SNR values indicate stronger signal-to-noise ratios, which translate to higher data rates and fewer retransmissions. The PCB sensor shows the highest SNR values, while LEWIS1 shows the lowest SNR values. Notably, the SNR values of LEWIS1-β are, on average, 23% higher than those of LEWIS1, signifying that LEWIS1-β produces responses with lower levels of noise compared to LEWIS1.

It is worth mentioning that SNR can be enhanced by reducing noise, which may originate from various sources, including external (electromagnetic), conducted, and intrinsic noise [32]. While conducted and external noise can be reduced through proper setup and circuit designs, intrinsic noise, although not eliminable, can be minimized through suitable heater and thermal sensor element design. In both the time and frequency domains, the accelerations recorded by the LEWIS accelerometers and those by the PCB accelerometer exhibited good agreement. It should be emphasized here that the aim of this research is not necessarily to develop sensors that outperform commercially available devices but rather to develop reasonably reliable, accurate, and cost-effective sensors compared to commercial equipment.

## 6. Educational Activities

The researchers at the University of New Mexico’s Smart Management of Infrastructure Laboratory (SMILab) have developed sensor classes and workshops for students, engineers, and educators since 2015. They used the LEWIS1 and LEWIS1-β sensors as educational tools for diverse participant groups involved in outreach activities and educational programs aimed at sensing and measuring the vibration of their surrounding area. These groups include a wide range of participants such as pre-K-12th students, college students, engineering students, and professionals from various fields who participated in sensor classes, workshops, or outreach activities. The purpose of these activities was to learn how to fabricate a low-cost sensor such as the LEWIS1 and to improve hands-on technical skills. And then, employ that sensor in real-world applications. This will boost the individual’s use of research and increase their creativity as they can think of how this sensor can be used in different areas. Additionally, the K-12th students and learners of different ages have the opportunity to experience exposure to research and industry applications and also think about integrating new technologies with existing infrastructure.

The total number of participants that attended and built the sensor from 2015 to fall 2023 are 439 and 241, respectively. Figure 12 indicates the percentage of different groups including elementary school, middle school, high school, and college/university students, and also the professionals from different fields who attended the sensor-building workshops. Half of the attendees were college/university students with different majors. High school, professionals, middle school, and elementary schools participated 21%, 15%, 10%, and 4%, respectively. The number of sensors and attendees participating in sensor workshops has an ascending trend except for the years 2019, 2020, and 2021. In these three mentioned years, the UNM team was not able to hold the in-person sensor workshop due to COVID; however, there were online classes for students where they had the sensor kit in their home and the instructor taught them the fabrication process online.

These classes and workshops have been developed as educational initiatives, aimed at acquainting civil engineering students with sensor technology and the principles of structural health monitoring. The final objective is to provide a comprehensive understanding of measurement concepts and their practical application within this field. This linkage to real-world application is intended to enrich the educational experience, inspiring students to find solutions that could be beneficial in civil engineering.

Figure 13a shows students using the LEWIS sensor for the purpose of measuring vibrations in a steel frame in the year 2015. This marks the initial stage when the curriculum incorporated sensor fabrication, complemented by practical applications in the domain of vibration measurement. In Figure 13b, we observe UNM civil engineering students as they were using their fabricated LEWIS1 sensor and conducting testing on the Arroyo Del Oso Bridge located in New Mexico, in 2022. These students accomplished modal analysis by deploying eight LEWIS sensors on the bridge structure.

## 7. Conclusions

Sensors are one of the useful technology tools that can be used in monitoring structures and education. While it is essential for engineers to gather data in order to have sufficient information to make decisions, there has not been much research conducted on the fabrication of low-cost sensors. Therefore, this paper introduced a low-cost tri-accelerometer sensor called LEWIS1 capable of gathering experimental and field data measurements. Additionally, an upgraded version of this sensor, LEWIS1-β, was developed that can measure the tri-axis acceleration, rotation, and temperature with a low-cost, portable, small-size, and user-friendly system. These two sensors were validated for data accuracy against a commercial PCB sensor using three different excitation types. The results show that the LEWIS1-β sensor performs comparably to the expensive commercial accelerometer and is more accurate than the LEWIS1 sensor with a similar cost. LEWIS1-β offers the advantage of maintaining a simple design, allowing engineers and researchers with limited knowledge of sensor technology to quickly fabricate and use the sensor for structural response measurements. This suggests that low-cost sensors can be used effectively for structural health monitoring, providing civil engineers with the means to fabricate and employ sensors tailored to their needs. Moreover, to improve the application and usability in SHM, Arduino wireless shield can be incorporated, along with a battery and solar panel, providing new opportunities for researchers and engineers.

The authors also have used this sensor to implement different educational activities, workshops, and sensor classes aimed at training and educational purposes since 2015. The LEWIS1 sensor is quite simple to fabricate, easy to be programmed, and can be used for different applications. Therefore, the researcher utilized the LEWIS1 sensor in outreach projects, engaging K-12 students, college students, civil engineering students, and professionals from various fields in sensor-building workshops. These workshops enabled participants to sense and quantify environmental vibrations and integrate the sensor into their daily lives.

## Figures and Tables

**Figure 1 sensors-24-05308-f001:**
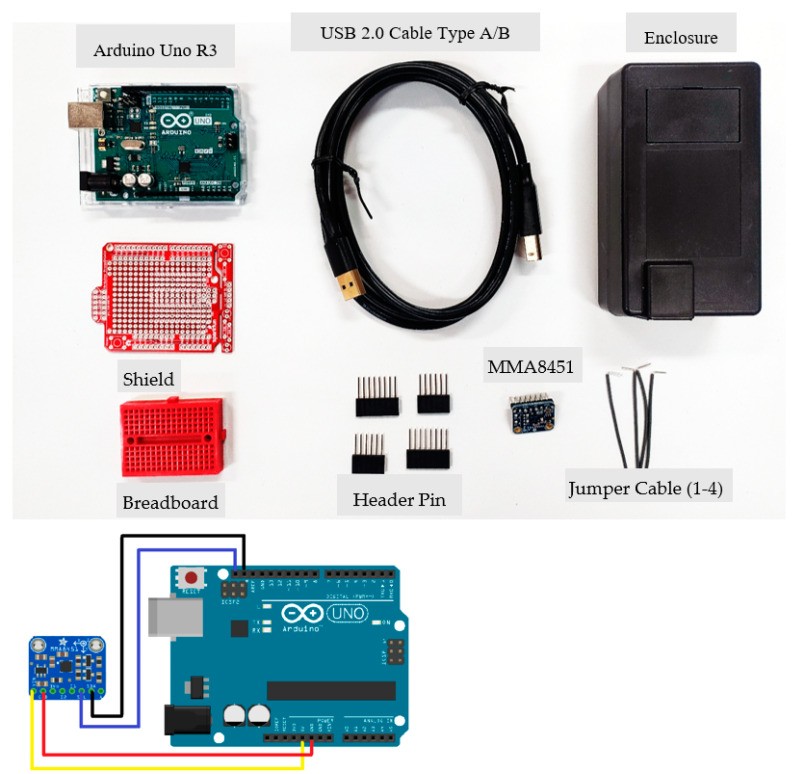
LEWIS1 component and wiring.

**Figure 2 sensors-24-05308-f002:**
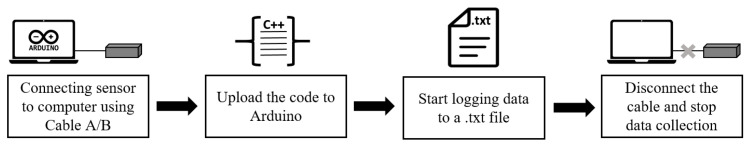
LEWIS1 sensor data collection.

**Figure 3 sensors-24-05308-f003:**
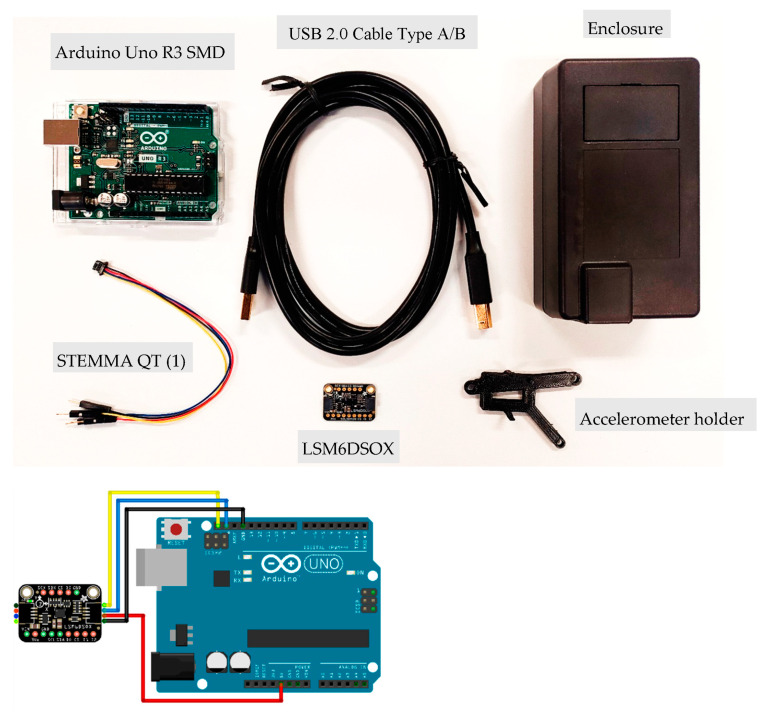
LEWIS1-β component.

**Figure 4 sensors-24-05308-f004:**
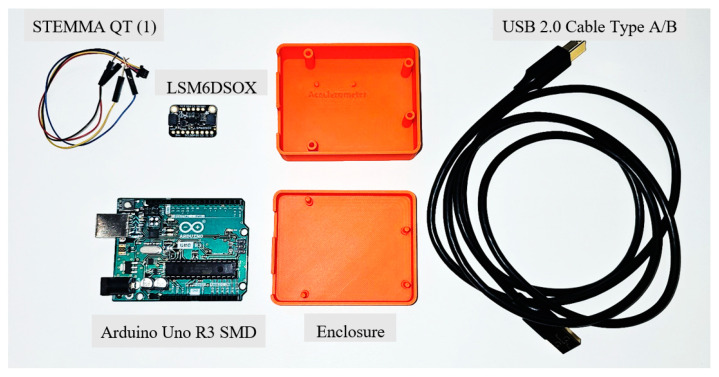
LEWIS1-γ component.

**Figure 5 sensors-24-05308-f005:**
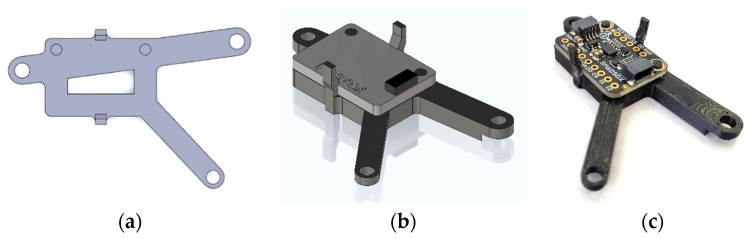
LEWIS1-β 3D printed holder: (**a**) top view; (**b**) isometric view; and (**c**) photo.

**Figure 6 sensors-24-05308-f006:**
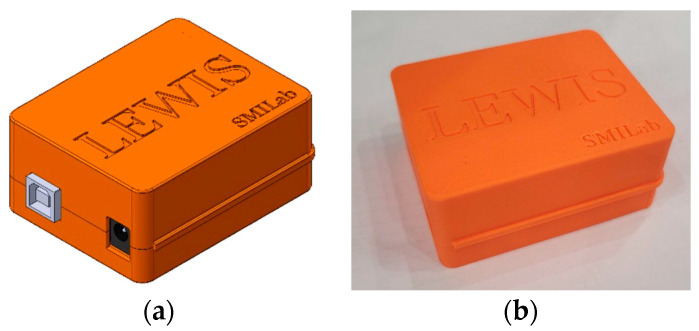
LEWIS1-γ 3D printed enclosure: (**a**) top view; (**b**) photo.

**Figure 7 sensors-24-05308-f007:**
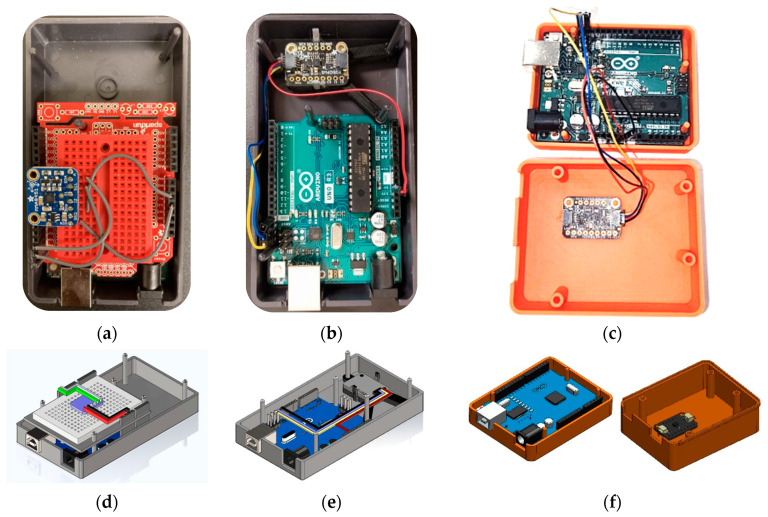
Plan view comparison: (**a**) LEWIS1 real photo, (**b**) LEWIS1-β real photo, (**c**) LEWIS1-γ real photo, (**d**) LEWIS1 3D model, (**e**) LEWIS1-β 3D model, and (**f**) LEWIS1-γ 3D model.

**Figure 8 sensors-24-05308-f008:**
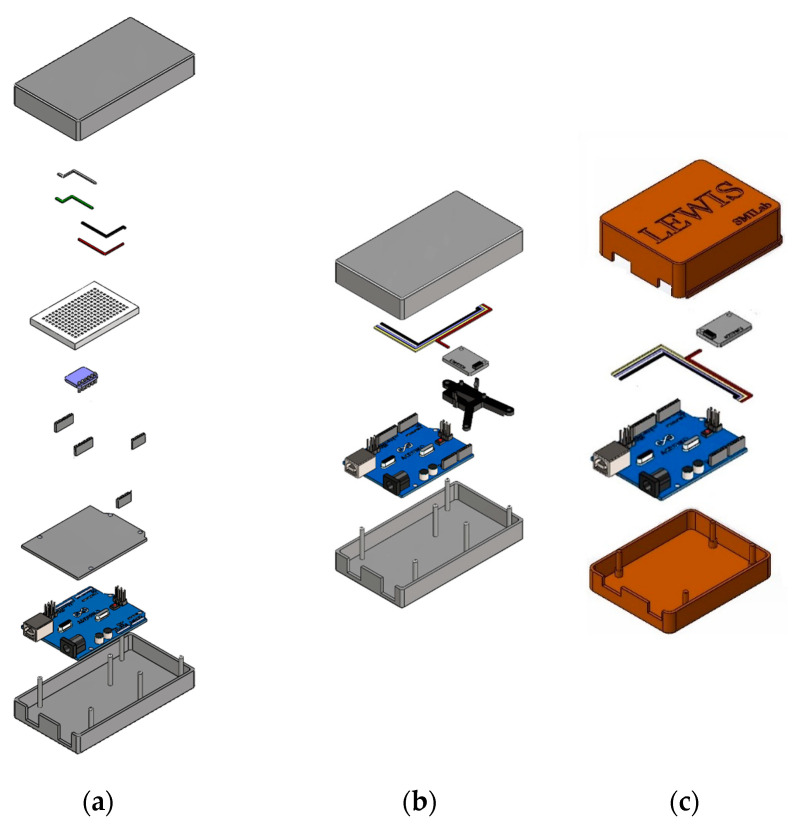
Number of components: (**a**) LEWIS1; (**b**) LEWIS1-β; and (**c**) LEWIS1-γ.

**Figure 9 sensors-24-05308-f009:**
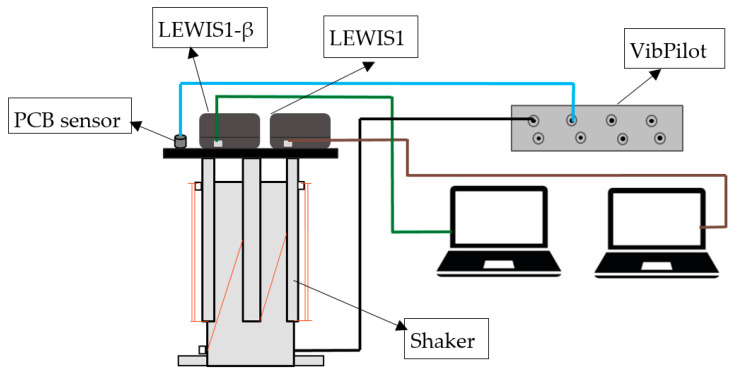
Experiment data collection platform.

**Figure 10 sensors-24-05308-f010:**
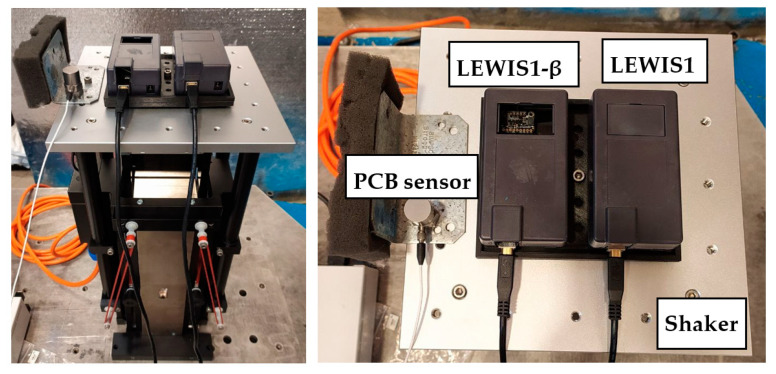
Experiment setup.

**Figure 11 sensors-24-05308-f011:**
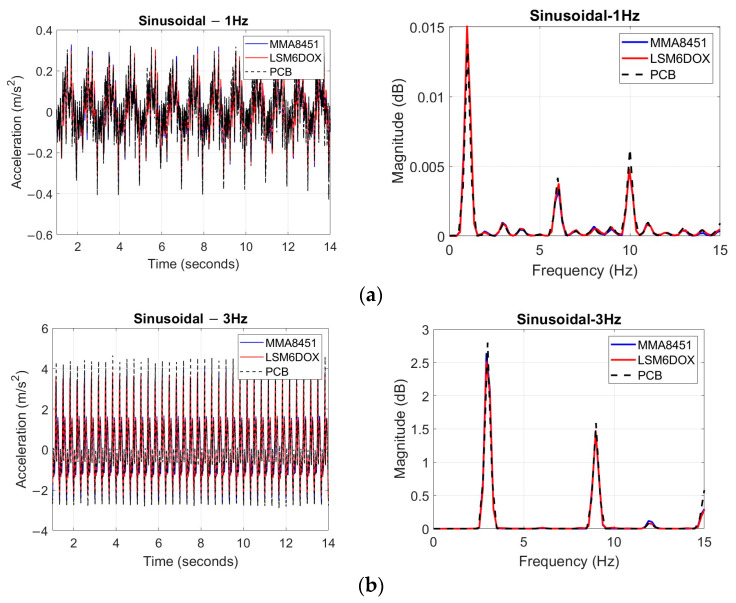
Time domain and frequency domain response: (**a**) sinusoidal 1 Hz; (**b**) sinusoidal 3 Hz; (**c**) sinusoidal 5 Hz; (**d**) sinusoidal 10 Hz; (**e**) band-limited white noise; and (**f**) sine sweep.

**Figure 12 sensors-24-05308-f012:**
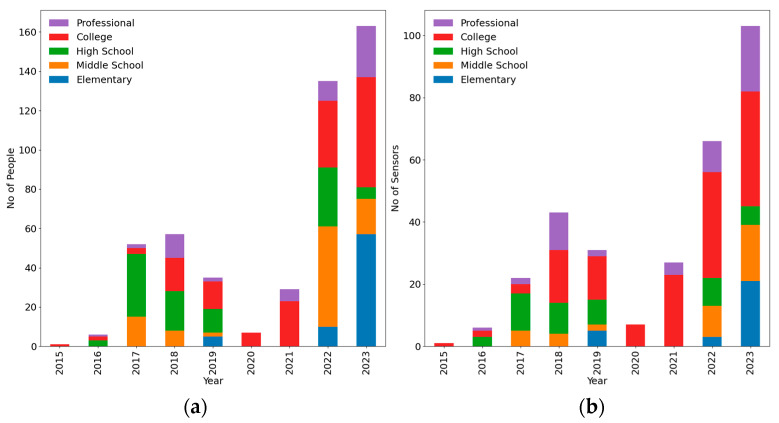
Distribution of different groups of attendees from 2015 to 2023: (**a**) No. of people; (**b**) No. of sensor.

**Figure 13 sensors-24-05308-f013:**
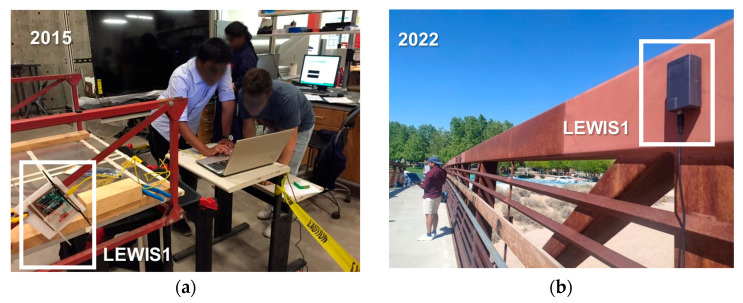
Experiment with LEWIS sensor: (**a**) vibration of steel structural frame—2015; (**b**) modal analysis of real bridge—2022.

**Table 1 sensors-24-05308-t001:** Cost of LEWIS1 components.

Component	Price, USD
Arduino Uno R3	USD 25.30
MMA8451 Accelerometers	USD 7.95
Breadboard and shield (Adafruit LLC, New York City, NY, USA)	USD 2.12
USB 2.0 Cable Type A/B (Adafruit LLC, New York City, NY, USA)	USD 7.00
Enclosure (Adafruit LLC, New York City, NY, USA)	USD 8.81
Total	USD 51.81

**Table 2 sensors-24-05308-t002:** Cost of LEWIS1-β and LEWIS-Υ components.

Component	LEWIS1-β Price, USD	LEWIS1-γ Price, USD
Arduino Uno R3 SMD	USD 24.10	USD 24.10
LSM6DSOX Accelerometers	USD 11.95	USD 11.95
STEMMA QT/Qwiic JST SH 4-pin to Premium Male Headers Cable (Adafruit LLC, New York City, NY, USA)	USD 0.95	USD 0.95
USB 2.0 Cable Type A/B	USD 7.00	USD 7.00
Enclosure	USD 8.81	USD 2.00
Total	USD 52.81	USD 46.00

**Table 3 sensors-24-05308-t003:** LEWIS1, LEWIS1-β, and LEWIS1-γ required number of components.

Component	LEWIS1	LEWIS1-β	LEWIS1-γ
Arduino board	1	1	1
Accelerometer	1	1	1
Accelerometer holder	0	1	0
Wire connection	4 to 4	1 to 4	1 to 4
Breadboard	1	0	0
Shield	1	0	0
Header pin	4	0	0
USB cable	1	1	1
Enclosure	1	1	1
Total	14	6	5

**Table 4 sensors-24-05308-t004:** Performance and feature comparison of LEWIS1, LEWIS1-β, and LEWIS1-γ sensors.

Properties	LEWIS1	LEWIS1-β	LEWIS1-γ	Improved?
Accelerometer	MMA8451	LSM6DOX	LSM6DOX	-
No. of components	14	6	5	✓
ADC	14 bit	16 bit	16 bit	✓
Accelerometer range	±2 g/±4 g/±8 g (±2 g for SHM)	±2 g/±4 g/±8 g/±16 g (±2 g for SHM)	±2 g/±4 g/±8 g/±16 g (±2 g for SHM)	✓
Gyroscope range	-	±125/±250/±500/±1000/±2000 dps (±250 dps for SHM)	±125/±250/±500/±1000/±2000 dps (±250 dps for SHM)	✓
Fabrication time	20 min	10 min	10 min	✓
Cost	USD 51.18	USD 52.81	USD 46.00	✓
Special requirement	Soldering the header pin to accelerometer	3D printing the holder	3D printing the box	-
Application	3DOF Accelerometer	3DOF Accelerometer, 3DOF Gyroscope, Temperature	3DOF Accelerometer, 3DOF Gyroscope, Temperature	✓

**Table 5 sensors-24-05308-t005:** LEWIS1, LEWIS1-β, and LEWIS1-γ requirements in SHM application.

Requirement	Value/Domain	Specification for SHM	LEWIS1	LEWIS1-β	LEWIS1-γ
Acceleration range	±2 g	Capable for global and local SHM application—component and structure level	2	3	3
Resolution	14 bit	SHM signal processing resolution including signal-to-noise ratio and ambient vibration requirements	2	3	3
Sampling rate	100 Hz	General SHM recommendation: 10 times the highest natural frequency of interest. For example: 10 Hz for bridges, 30 Hz for buildings.	1	3	3
Validated in real applications to investigate SHM standards for specific infrastructure applications, to identify value for SHM	Railway bridge	Monitoring vibration levels following AREMA Manual for Railway Engineering for bridge vibration [29]	3	1	1
	Highway bridge	Monitoring vibration levels to explore with AASHTO LRFD Bridge Design Specifications [30]	1	1	1
	Pedestrian bridge	Monitoring vibration levels in compliance with ISO 10137 [31]	3	3	1
Installation for SHM campaigns and attachment to structures	Flexible, easy, and firm installation for SHM applications	Based on deployment strategy and preferably on critical load-bearing components	1	2	2
Orientation	Flexible orientation placement, easy to confirm orientation	Three-axis accelerometer for all models; additional three-axis gyroscope for LEWIS1-β and LEWIS1-γ to measure rotational movements and torsion.	1	3	3
Total SHM score			14	19	17

**Table 6 sensors-24-05308-t006:** Excitation specification.

Type of Excitation	Frequency (Hz)
Sinusoidal	1
	3
	5
	10
Sine sweep	0–10
Band-limited white noise	10

**Table 7 sensors-24-05308-t007:** RMSE and SNR of MMA8451, LSM6DOX, and PCB.

Type of Excitation	Frequency (Hz)	RMSE		Difference (%)	SNR		
		LEWIS1	LEWIS1-β		LEWIS1	LEWIS1-β	PCB
Sinusoidal	1	0.52	0.52	0.77	8.68	9.63	9.87
	3	0.16	0.10	42.22	10.89	12.10	26.11
	5	3.17	1.85	52.90	12.48	14.66	27.20
	10	2.61	1.26	69.82	16.54	25.14	30.48
Sine sweep	0–10	1.24	0.57	73.39	14.87	18.06	31.36
Band-limited white noise	10	3.11	1.52	68.96	22.05	29.27	34.70

## Data Availability

All data, models, or code that support the findings of this study are available from the corresponding author upon reasonable request.

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
