# Peer review of "Low-Cost Efficient Wireless Intelligent Sensor (LEWIS) for Research and Education"

_sensors, 2024, doi:10.3390/s24165308_

Round 1
Reviewer 1 Report
Comments and Suggestions for Authors
The work presents an interesting system of instruction for a SHM, encompassing a commendable range of educational activities. I extend my congratulations to the authors for their great efforts. The validation process using the shaker is particularly noteworthy. However, I believe there is room for enhancement in the project description of the LEWIS family, providing complementary information that would benefit users interested in this application. Below are my suggestions:
Section 2 appears to be rather brief. It would be beneficial to include a schematic illustrating the traditional wired monitoring system alongside your system, highlighting its advantages and associated costs. This section could be more thoroughly explored. If I understand correctly, your system is also wired, but at a lower cost and more modern as it concentrates everything on the Arduino.
Another recommendation for this section is to create a comprehensive set of requirements (you mention software requirements in line 145). These should be presented in a table, following established literature on how to formulate requirements. This would assist future designers and potential LEWIS users, justifying your design decisions and choice of architectures (you mention a simplistic architecture for LEWIS in line 1141, but it is not shown). This represents a gap in your work that should be addressed. The architecture of your system is fundamental for other engineers to understand your project. While Figure 8 provides a general diagram of LEWIS in an experiment, the architecture itself is not presented. I recommend reviewing (no need to cite) https://doi.org/10.1016/j.asr.2022.04.058 for a general line to describe your system to the reader.
In the paragraph starting on line 148, the hardware is presented briefly, but without the architecture, it becomes challenging for potential users, particularly those without advanced engineering knowledge. This recommendation applies to all three versions of LEWIS.
Just to clarify, does LEWIS write data to a memory card, or must it be connected via USB continuously? Adding a battery, solar panel, and memory card would take your system to the next level of possibilities.
I suggest including a comparative table of the sensor characteristics used in each version. For instance, Table 4 lists ranges from 2g to 16g with different gyro rates, but it is unclear which specific ones are used for SHM. You do not need 16g or 2000 deg/s.
Regarding LEWIS software, the section lacks sufficient detail, especially for users without advanced knowledge. Although I have extensive experience with this type of hardware, I would find it challenging to reproduce your system due to the lack of basic information. Perhaps consider making your code available on GitHub for those interested in this application.
When discussing validation, I recommend including a comparative plot of vibration accelerometer data in the frequency domain, as it is a common analysis method in this field.
Line 88: CHEAP (?)
Reviewer 2 Report
Comments and Suggestions for Authors
Thank you for inviting me to evaluate the paper “Low-cost Efficient Wireless Intelligent Sensor (LEWIS) for Engineering, Research, and Education”. The paper presented research on the application of LEWIS sensors for research and education activities. The paper needs some modification before consideration for publication in the esteemed journal based on following reasons.
1. Section 2. “Challenges and Benefits of Sensor Technology in SHM” should be combined into Section 1 “Introduction” since the content is somewhat repeated.
2. In Section 3, only sensor versions, which were experimentally verified compared to other sensors, are presented.
3. Please check the title of Section 3.1 (repeated three times)
4. The evaluation metrics (i.e., in Eqs. (1), (2), and (3)) should be relocated in Section of proposed sensors (these metrics put after proposing sensors).
5. The paper’s logic is somewhat complicated for readers. Please consider the outline as follows: I) Introduction; II) Prototype design of LEWIS1 (only one type of sensor presented); III) Experimental evaluation of the proposed sensor (using in educational activities with detailed analysis: signal, features of signals…); IV) Improvement of Current version (2nd version) with details of design and tested signals…V) Achievements and Challenges (of Proposed Sensors); VI) Conclusion.
6. As presented, the proposed sensors can measure up to three directional vibration signals. In Figure 10, which are vibration signals measured? And how about the other direction?
7. Currently, LEWIS sensors are used for education, research activities. For engineering, on-site test on real structures with detailed analysis has not been achieved. Please the paper’s title could be reconsidered.
Round 2
Reviewer 1 Report
Comments and Suggestions for Authors
I see that in this version the work has improved substantially and I congratulate the authors on the paper. I still have one specific recommendation that I think is fundamental to the work. At the beginning of section 3 I think the authors should include a table of requirements that are the premise of the LEWIS family. I'll put an example below of what I think these requirements should look like and you can use this as a basis and ideally even expand on it.
The SHM shall withstand a maximum vibration of XX g.
The SHM shall have a minimum resolution of XX bits for data recording.
The SHM shall provide a minimum sampling rate of XX Hz.
The SHM shall be capable of monitoring vibration levels in compliance with the standards specified for constructions, according to standard XX.
The SHM shall be installable at any point within the building or only at specific points as required.
The SHM shall be installed in the position/orientation XX.
Reviewer 2 Report
Comments and Suggestions for Authors
The paper has been edited according to the reviewers’ comments. It can be accepted for publication in the journal.
Author Response
We appreciate all your constructive feedback in enhancing the quality of this paper. Thanks.